# Antitumour activity of neratinib in patients with *HER2*-mutant advanced biliary tract cancers

James J. Harding [1,2] ✉, Sarina A. Piha-Paul[3], Ronak H. Shah [1,4], Jessica J. Murphy[1,4], James M. Cleary[5], Geoffrey I. Shapiro [5], David I. Quinn [6], Irene Braña[7,8], Victor Moreno [9], Mitesh Borad[10], Sherene Loi [11], Iben Spanggaard [12], Haeseong Park[13], James M. Ford [14], Mónica Arnedos[15], Salomon M. Stemmer[16,17], Christelle de la Fouchardiere [18], Christos Fountzilas [19,20], Jie Zhang[21], Daniel DiPrimeo[21], Casey Savin[1,4], S. Duygu Selcuklu[1,4], Michael F. Berger[1,4], Lisa D. Eli[21], Funda Meric-Bernstam [3], Komal Jhaveri[1,2], David B. Solit [1,2,4] & Ghassan K. Abou-Alfa[1,2]

*HER2* mutations are infrequent genomic events in biliary tract cancers (BTCs). Neratinib, an irreversible, pan-HER, oral tyrosine kinase inhibitor, interferes with constitutive receptor kinase activation and has activity in *HER2*-mutant tumours. SUMMIT is an open-label, single-arm, multi-cohort, phase 2, 'basket' trial of neratinib in patients with solid tumours harbouring oncogenic *HER2* somatic mutations (ClinicalTrials.gov: NCT01953926). The primary objective of the BTC cohort, which is now complete, is first objective response rate (ORR) to neratinib 240 mg orally daily. Secondary objectives include confirmed ORR, clinical benefit rate, progression-free survival, duration of response, overall survival, safety and tolerability. Genomic analyses were exploratory. Among 25 treatment-refractory patients (11 cholangiocarcinoma, 10 gallbladder, 4 ampullary cancers), the ORR is 16% (95% CI 4.5–36.1%). The most common *HER2* mutations are S310F (n = 11; 48%) and V777L (n = 4; 17%). Outcomes appear worse for ampullary tumours or those with co-occurring oncogenic *TP53* and *CDKN2A* alterations. Loss of amplified *HER2* S310F and acquisition of multiple previously undetected oncogenic co-mutations are identified at progression in one responder. Diarrhoea is the most common adverse event, with any-grade diarrhoea in 14 patients (56%). Although neratinib demonstrates antitumour activity in patients with refractory BTC harbouring *HER2* mutations, the primary endpoint was not met and combinations may be explored.

Biliary tract cancers (BTCs) represent an uncommon group of neoplasia that comprise tumours of the intrahepatic and extrahepatic biliary tree, gallbladder, and ampulla of Vater[1,2]. For over a decade, gemcitabine plus cisplatin has been an established first-line systemic treatment for patients with locally advanced/metastatic disease, although recent data indicate that the addition of anti-programmed cell death ligand-1 (PD-L1) therapy further improves outcomes[3,4]. The combination of gemcitabine and cisplatin plus durvalumab leads to an objective response rate (ORR) of 26.7%, progression-free survival (PFS) of 7.2 months, and overall survival (OS) of 12.8 months. In the second-line setting in

genomically unselected populations, fluoropyrimidine doublets exhibit anticancer activity and modestly improve patient outcomes versus monotherapy or placebo, with response rates of 5% and 15% for FOLFOX and liposomal irinotecan plus fluorouracil and leucovorin, respectively[5,6]. Implementation of precision medicine is an increasingly relevant strategy in the second-line setting, given the high proportion of druggable alterations identified in tumours of the biliary tree[7–10].

HER2 is a receptor tyrosine kinase encoded by the HER2 (ERBB2) gene[11]. HER2 protein overexpression, gene amplification, and less commonly, somatic HER2 mutations (ie, kinase domain missense and insertion mutations, extracellular domain missense mutations, and transmembrane domain mutations), drive uncontrolled cellular signalling, promoting tumour growth and survival[11]. Pharmacological inhibition of HER2 signalling has antitumour activity in preclinical models and is a validated therapeutic strategy in HER2-positive breast, gastric, and lung cancers[11].

HER2 alterations have been identified in a subset of BTCs and somatic HER2 mutations have been reported at frequencies of up to 10% in this setting[7,9,12]. HER2 alterations (amplifications or mutations) were associated with poor overall survival (OS) in patients with metastatic disease in a retrospective dataset[9] and HER2 overexpression was associated with increased risk of disease recurrence in patients with resected BTC[13]. Case reports, case series, and single-arm prospective studies suggest that targeting HER2 has therapeutic potential in patients with HER2-altered BTC; however, published data are limited by retrospective designs, small sample sizes, and in some cases lack of comprehensive genomic annotation[14–17]. Published studies have focused on targeting HER2-amplified or HER2-overexpressing tumours[18] and few prospective studies have sought to target HER2 in BTC harbouring activating somatic HER2 mutations[14,15,17,19].

Neratinib, an irreversible pan-HER oral tyrosine kinase inhibitor (TKI), interferes with constitutive receptor kinase activation, leading to cancer regression in preclinical models[20–22]. In the clinic, neratinib extends OS of patients with early- and late-stage HER2-positive breast cancer and is approved by the United States Food and Drug Administration for use as monotherapy in adjuvant breast cancer following 1 year of trastuzumab, and in combination with capecitabine in third-line metastatic breast cancer[23,24]. SUMMIT was an open-label, international phase 2 'basket' trial investigating the activity and safety of neratinib across a broad spectrum of cancers in patients whose tumours harbour activating somatic HER2 mutations[25]. In the initial study report, the antitumour activity of neratinib appeared to be dependent on both histology and mutation. One of the first seven patients enrolled in the HER2-mutant BTC cohort in SUMMIT achieved a partial response (PR), meeting Simon two-stage criteria for cohort expansion[25]. Here, we report the final results of the expanded HER2-mutant BTC cohort in SUMMIT.

## Results

Between April 3, 2014, and August 1, 2019, 25 patients with metastatic BTC harbouring HER2 mutations were enrolled at hospitals in the USA, Australia, Denmark, France, Israel, and Spain, and treated on study (Fig. 1).

Patient characteristics are summarised in Table 1.

At data cut-off (January 22, 2021), all 25 patients (100%) had discontinued treatment; at the end of study, 19 (76%) had died of disease, four (16%) had withdrawn consent for additional follow-up, and 2 (8%) were ongoing survival follow-up. All patients received at least one dose of study drug. Median time on treatment was 6.7 (interquartile range [IQR] 4.0–16.4) months for the overall cohort; median follow-up duration was 9.0 (IQR 3.7–18.4) months.

### Efficacy

Six of the 25 patients with BTC discontinued the study because of clinical deterioration or clinical progression and were not evaluable for

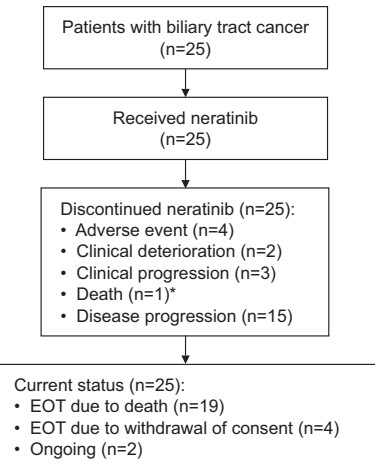

**Fig. 1 | Study flow.** *Death was due to progressive disease. *EOS* end of study.

**Table 1 | Baseline demographics and clinical characteristics**

|  | *HER2*-mutant biliary cohort (*n* = 25) |
|---|---|
| Age, years | 65 (49–78) |
| Female sex | 13 (52%) |
| **ECOG performance status** |  |
| 0 | 6 (24%) |
| 1 | 17 (68%) |
| 2 | 2 (8%) |
| **Tumour site** |  |
| Cholangiocarcinoma | 11 (44%) |
| Intrahepatic | 6 (24%) |
| Extrahepatic | 5 (20%) |
| Gallbladder | 10 (40%) |
| Ampulla of Vater[a] | 4 (16%) |
| **Histology[46]** |  |
| Adenocarcinoma | 22 (88%) |
| Well differentiated | 2 (8%) |
| Moderately differentiated | 9 (36%) |
| Poorly differentiated | 7 (28%) |
| Unknown | 4 (16%) |
| Other | 3 (12%) |
| **M category at enrolment** |  |
| M0 | 1 (4%) |
| M1 | 24 (96%) |
| Patients with prior surgery | 16 (64%) |
| Patients with prior radiation | 5 (20%) |
| Prior systemic regimens | 2 (0–7) |
| **Prior systemic therapy** |  |
| Gemcitabine-based | 24 (96%) |
| Platinum-based | 23 (92%) |
| Fluoropyrimidine-based | 18 (72%) |
| None | 1 (4%) |

Values are median (range) or *n* (%) unless otherwise indicated. *ECOG* Eastern Cooperative Oncology Group. Data cut-off: Jan 22, 2021.
[a]One of four ampullary cancers had intestinal morphology.

response. Among the remaining 19, four had a confirmed PR (Fig. 2a; Table 2), for a confirmed objective response rate (ORR) of 16% (95% confidence interval [CI] 4.5–36.1%). These confirmed PRs were observed in three patients with gallbladder carcinoma (3/10; 30%) and one with cholangiocarcinoma (1/11; 9%). Another patient with cholangiocarcinoma had an unconfirmed PR. None of four patients with

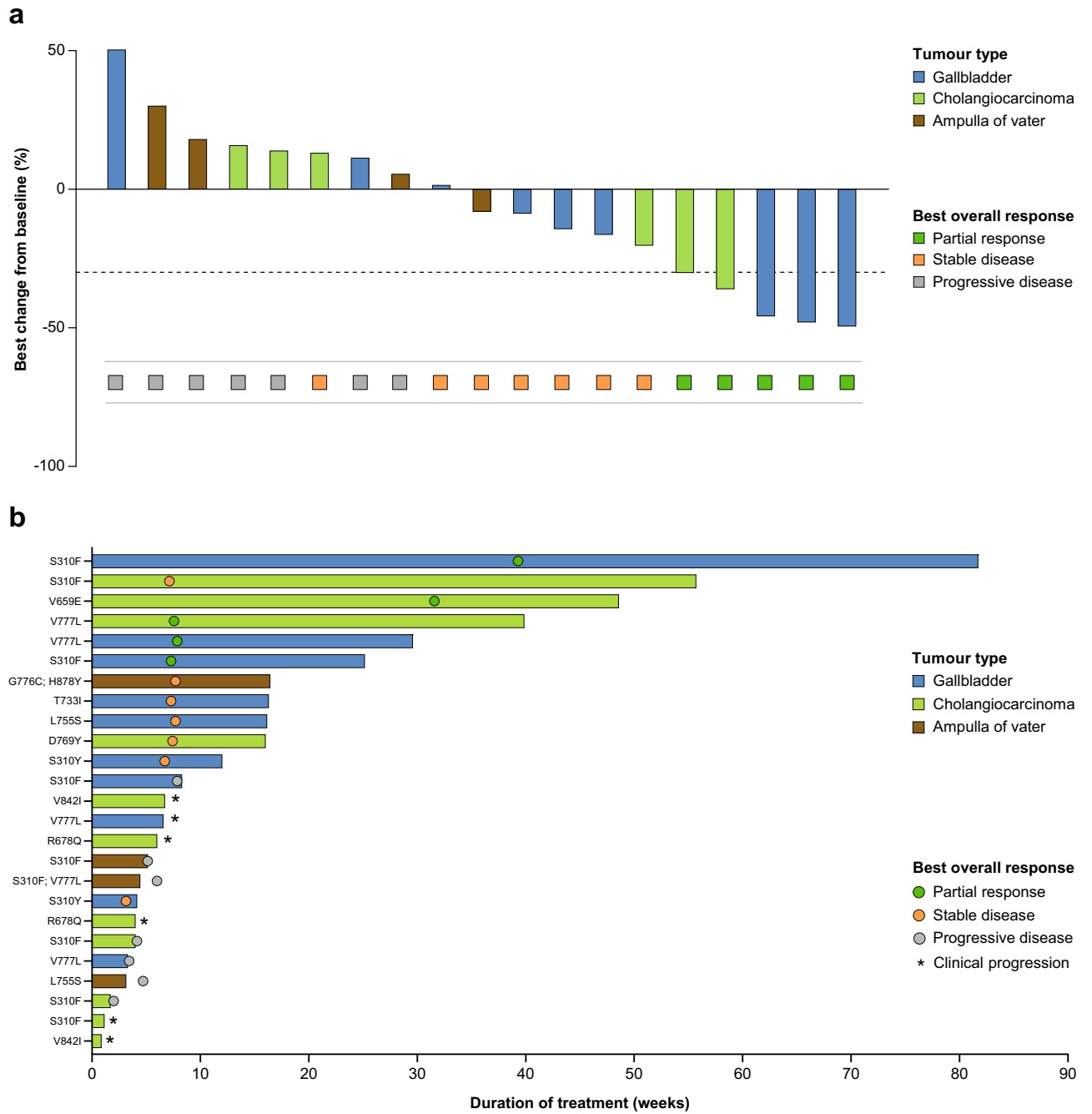

**Fig. 2 | Activity of treatment (*n* = 25). a** Waterfall plot for 19 patients with Response Evaluation Criteria in Solid Tumours (version 1.1)-evaluable disease; six patients who were not evaluable are not depicted. Dotted line indicates −30% tumor shrinkage; **b** Time on treatment and response assessment for all 25 study patients. Data cut-off: January 22, 2021. *CI* confidence interval. Source data are provided as a Source Data file.

cancer of the ampulla of Vater had a response. The duration of response (DoR) for the four patients with PR were 3.0, 3.7 (censored), 3.8, and 4.7 months (Fig. 2b). The best overall response (BOR) rate was 20% (95% CI 6.8–40.7%); the clinical benefit rate (CBR) was 28.0% (95% CI 12.1–49.4%), and the disease control rate (DCR; not a prespecified analysis) was 24.0% (95% CI 9.3–45.1%).

Median progression-free survival (PFS) was 2.8 (95% CI 1.1–3.7) months (Fig. 3a; Table 2); median PFS for the gallbladder, cholangiocarcinoma, and ampulla subsets were 3.7 (95% CI 0.8–6.4), 1.4 (95% CI 0.5–9.1), and 1.1 (95% CI 1.1–3.8) months, respectively. Median OS was 5.4 (95% CI 3.7–11.7) months (Fig. 3b; Table 2); median OS for the gallbladder, cholangiocarcinoma, and ampulla subsets were 9.8

(95% CI 2.4-not estimable), 5.4 (95% CI 0.8–16.2), and 5.0 (95% CI 3.7–10.2) months, respectively.

## Safety

All 25 patients in the BTC cohort had at least one adverse event (AE; Table 3); 16 (64%) had one or more serious AEs, two (8%) had serious treatment-related AEs (diarrhoea, dehydration, acute kidney injury; Supplementary Table 2), and five (20%) had treatment-emergent AEs and/or clinical progression leading to treatment discontinuation. Diarrhoea was the most common AE in the BTC cohort; 14 patients (56%) reported diarrhoea of any grade and six (24%) had a grade 3 diarrhoea event (Supplementary Table 3). There was no grade 4

diarrhoea. Two patients had a grade 5 AE: one died because of general deterioration and one because of sepsis.

## Exploratory genomic analysis

Twenty-three patients (92%) had either sufficient pre-treatment plasma-derived cell-free DNA (cfDNA), tumour tissue, or both, for retrospective central next-generation sequencing (NGS; Supplementary Table 1). Fifteen patients (60%) had adequate archival or pre-treatment tissue for NGS analysis, an additional eight (32%) had pre-treatment cfDNA. In one of 23 patients, a *HER2* V842I reported by enrolment assay was not identified on central NGS, and a custom single-gene assay was inconclusive. In three of 23 patients, central testing not only confirmed the *HER2* mutation reported on enrolment assay but also identified additional *HER2* mutations (D277Y, V842I,

L253V). In the remaining patients, *HER2* mutations detected by enrolment assay and central NGS were concordant.

Based on retrospective central NGS (23 of 25 patients), *HER2* mutations were distributed as shown in Fig. 4a. The most common mutation was S310F, an extracellular domain hotspot mutation (*n* = 11; 48%), followed by kinase domain hotspot mutation V777L (*n* = 4, 17%). Objective responses occurred in patients with S310F (*n* = 2), V659E (*n* = 1; unconfirmed PR), and V777L (*n* = 2) mutations. The most common co-occurring alterations included *TP53* (*n* = 13; 57%), *CDKN2A* (*n* = 5; 22%), *ERBB3* (*n* = 4; 17%, 2 oncogenic and 2 variants of unknown significance), *SMAD4* (*n* = 4; 17%), and *SKT11* (*n* = 4; 17%) (Fig. 4b). Two patients (9%) had tumours harbouring co-occurring *HER2* copy-

### Table 2 | Activity summary

| Activity endpoint[a] | *HER2*-mutant biliary cohort (*n* = 25) |
|---|---|
| Objective response at first assessment (week 8) | 2/18 (11.1%) |
| **Objective response (confirmed)[b]** | |
| CR | 0 |
| PR | 4 (16%) |
| ORR | 16.0% (4.5–36.1%) |
| BOR | 5 (20%) |
| DOR for each responder, months | 3.0, 3.6[c], 3.7, 4.7 |
| Median time on treatment (IQR), months | 6.7 (4.0–16.4) |
| CBR[d] | 28.0% (12.1–49.4%) |
| CR | 0 |
| PR | 4 (16%) |
| SD ≥ 16 weeks | 3 (12%) |
| Median PFS (95% CI), months[e] | 2.8 (1.1–3.7) |
| Median OS (95% CI), months | 5.4 (3.7–11.7) |

Data are *n* (%) or % (95% CI) unless otherwise indicated.

*CBR* clinical benefit rate, *CI* confidence interval, *CR* complete response, *DoR* duration of response, *IQR* interquartile range, *ORR* objective response rate, *OS* overall survival, *PFS* progression-free survival, *PR* partial response, *SD* stable disease.

[a]Response is based on investigator tumour assessments per Response Evaluation Criteria in Solid Tumours (version 1.1).

[b]Objective response rate is defined as either a complete or partial response that is confirmed no less than 4 weeks after the criteria for response are initially met.

[c]Censored.

[d]Clinical benefit rate is defined as confirmed complete or partial response or stable disease for at least 16 weeks (within ±7-day visit window).

[e]Kaplan–Meier analysis.

### Table 3 | Incidence of treatment-emergent adverse events (occurring in ≥10% of patients) (*n* = 25)

| Adverse event, *n* (%) | All grades | Grade 1/2 | Grade 3/4 |
|---|---|---|---|
| Diarrhoea[a] | 14 (56%) | 8 (32%) | 6 (24%)[b] |
| Vomiting | 12 (48%) | 11 (44%) | 1 (4%) |
| Fatigue | 10 (40%) | 10 (40%) | 0 |
| Nausea | 10 (40%) | 10 (40%) | 0 |
| Abdominal pain | 8 (32%) | 6 (24%) | 2 (8%) |
| Decreased appetite | 7 (28%) | 7 (28%) | 0 |
| Constipation | 6 (24%) | 6 (24%) | 0 |
| Aspartate aminotransferase increased | 4 (16%) | 3 (12%) | 1 (4%) |
| Dehydration | 4 (16%) | 2 (8%) | 2 (8%) |
| Dizziness | 4 (16%) | 4 (16%) | 0 |
| Dry mouth | 4 (16%) | 4 (16%) | 0 |
| Pyrexia | 4 (16%) | 4 (16%) | 0 |
| Abdominal distension | 3 (12%) | 3 (12%) | 0 |
| Anaemia | 3 (12%) | 2 (8%) | 1 (4%) |
| Ascites | 3 (12%) | 2 (8%) | 1 (4%) |
| Asthenia | 3 (12%) | 2 (8%) | 1 (4%) |
| Blood alkaline phosphatase increased | 3 (12%) | 1 (4%) | 2 (8%) |
| Blood bilirubin increased | 3 (12%) | 1 (4%) | 2 (8%) |
| Hypokalaemia | 3 (12%) | 3 (12%) | 0 |
| Rash | 3 (12%) | 3 (12%) | 0 |
| Weight decreased | 3 (12%) | 3 (12%) | 0 |

[a]None of the diarrhoea events resulted in dose discontinuation; one patient was hospitalised, and four patients reduced study drug due to diarrhoea events.

[b]No grade 4 diarrhoea events were reported.

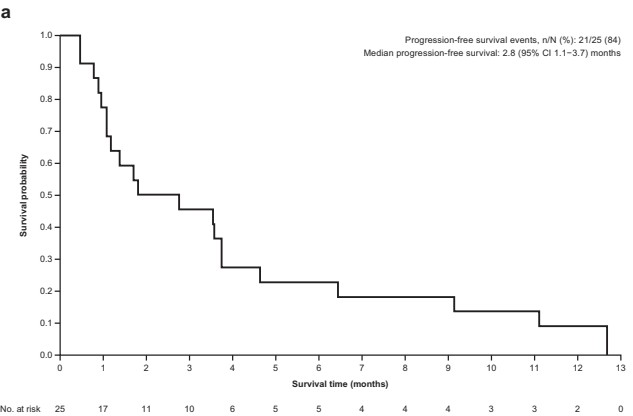

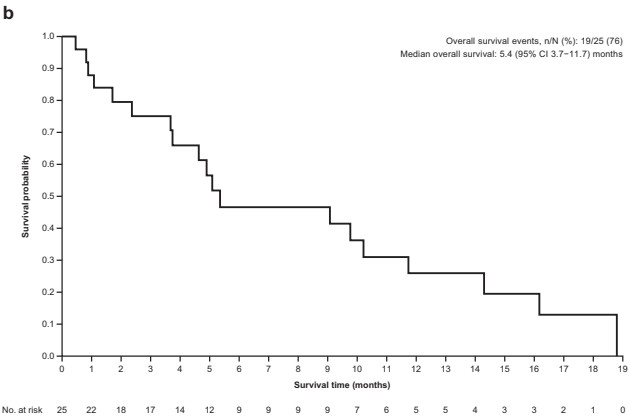

Progression-free survival events, n/N (%): 21/25 (84)
Median progression-free survival: 2.8 (95% CI 1.1–3.7) months

Overall survival events, n/N (%): 19/25 (76)
Median overall survival: 5.4 (95% CI 3.7–11.7) months

**Fig. 3 | Esimated Surival For HER2-mutant Biliary Tract Cancer Patients Treated with Neratenib.** Kaplan–Meier curves for **a** progression-free survival and **b** overall survival. Source data are provided as a Source Data file.

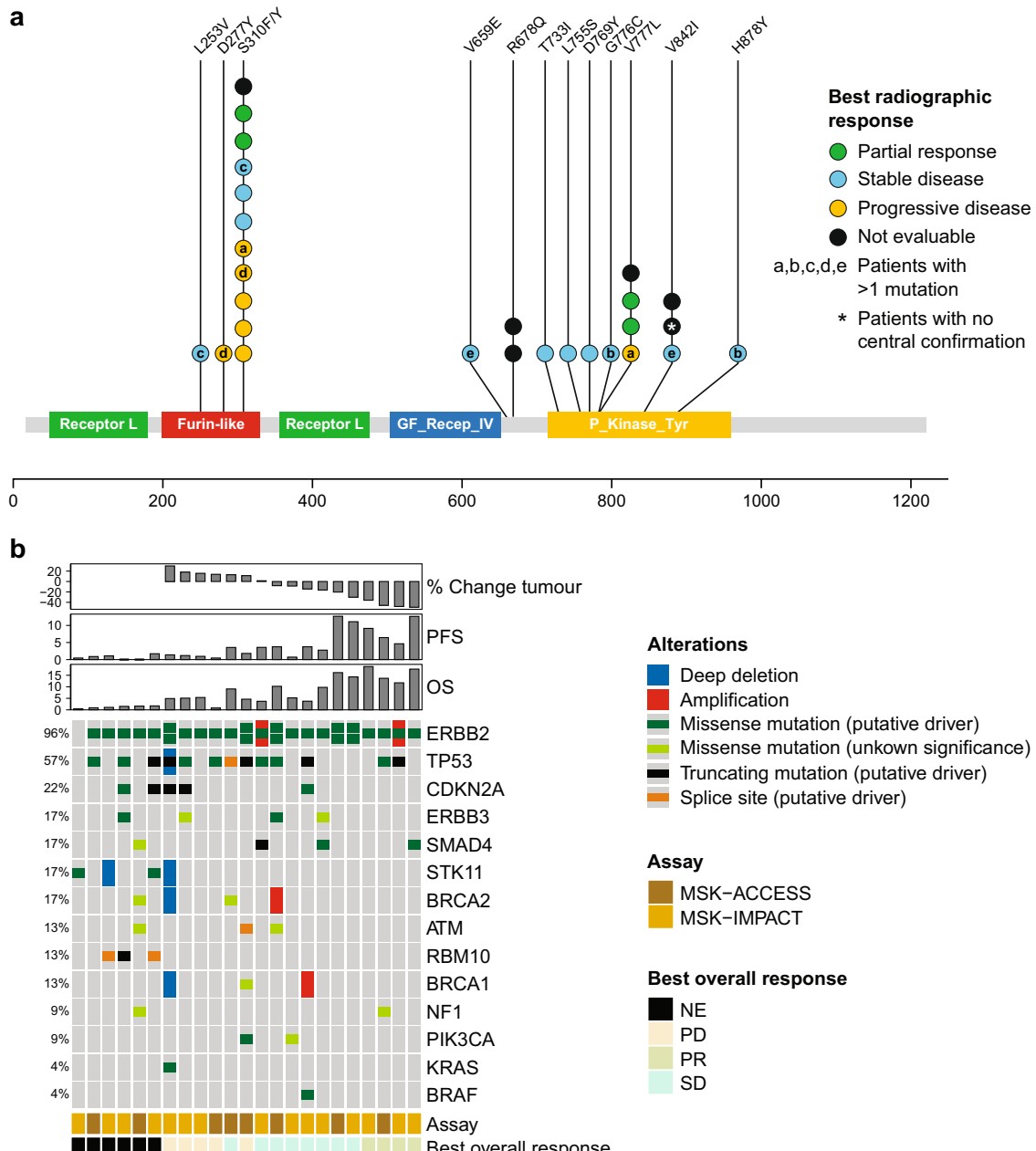

**Fig. 4 | Genomic determinates of response to neratinib (n = 23). a** Lollipop diagram of the *HER2* gene annotated with centrally confirmed mutations and tumour responses. Letters within circles indicate matching samples, i.e., patients with more than one mutation; **b** OncoPrint of co-occurring genomic alterations annotated with objective response, central next-generation sequencing confirmation assay best tumour shrinkage, PFS, and OS. NE not estimable, OS overall survival, PD progressive disease, PFS progression-free survival, PR partial response, SD stable disease. Source data are provided as a Source Data file.

number amplifications, one mutant allele and one wild-type allele. Two patients (9%) had tumours harbouring co-occurring *PIK3CA* mutations (one oncogenic E545K alteration and an E81K variant of unknown significance). Oncogenic MAPK pathway co-alterations included *KRAS* Q61H (n = 1; 4%) and *BRAF* D594N (n = 1; 4%). No *IDH1*, *EGFR*, or *FGFR2* alterations were observed.

As retrospective analysis suggested worse outcomes for those patients whose tumours harboured *TP53* and *CDK2NA* mutations, we descriptively explored potential association of these co-occurring mutations on outcome[26]. Among 10 patients with tumours wild type for both *TP53* and *CDK2NA*, two (20%) achieved a PR; the median OS for this group was 14.3 (95% CI 0.46–18.8) months. Two of eight patients with co-occurring *TP53* mutations without *CDK2NA* mutations (25%) achieved a PR; median OS was 9.1 (95% CI 0.82–11.7) months. There

were no PRs among five patients with co-occurring *TP53* and *CDK2NA* mutations; median OS for this group was 4.9 (95% CI 1.7–5.1) months. All *TP53* and *CDK2NA* mutations identified were either oncogenic or likely oncogenic according to OncoKB.

To explore hypothesis-generating changes in genomic profile coincident with progression on neratinib, paired tumour biopsies and serially collected cfDNA were interrogated by next-generation sequencing (NGS) in the four responders. One had paired pre-treatment and at-progression biopsies, and sequentially collected cfDNA during treatment. In the remaining three patients, before-treatment, on-treatment, and at-progression cfDNA samples were available for analysis. No genomic alterations were detected in cfDNA collected from two patients. One patient had adenosquamous carcinoma of the gallbladder harbouring a *HER2* S310F mutation whose

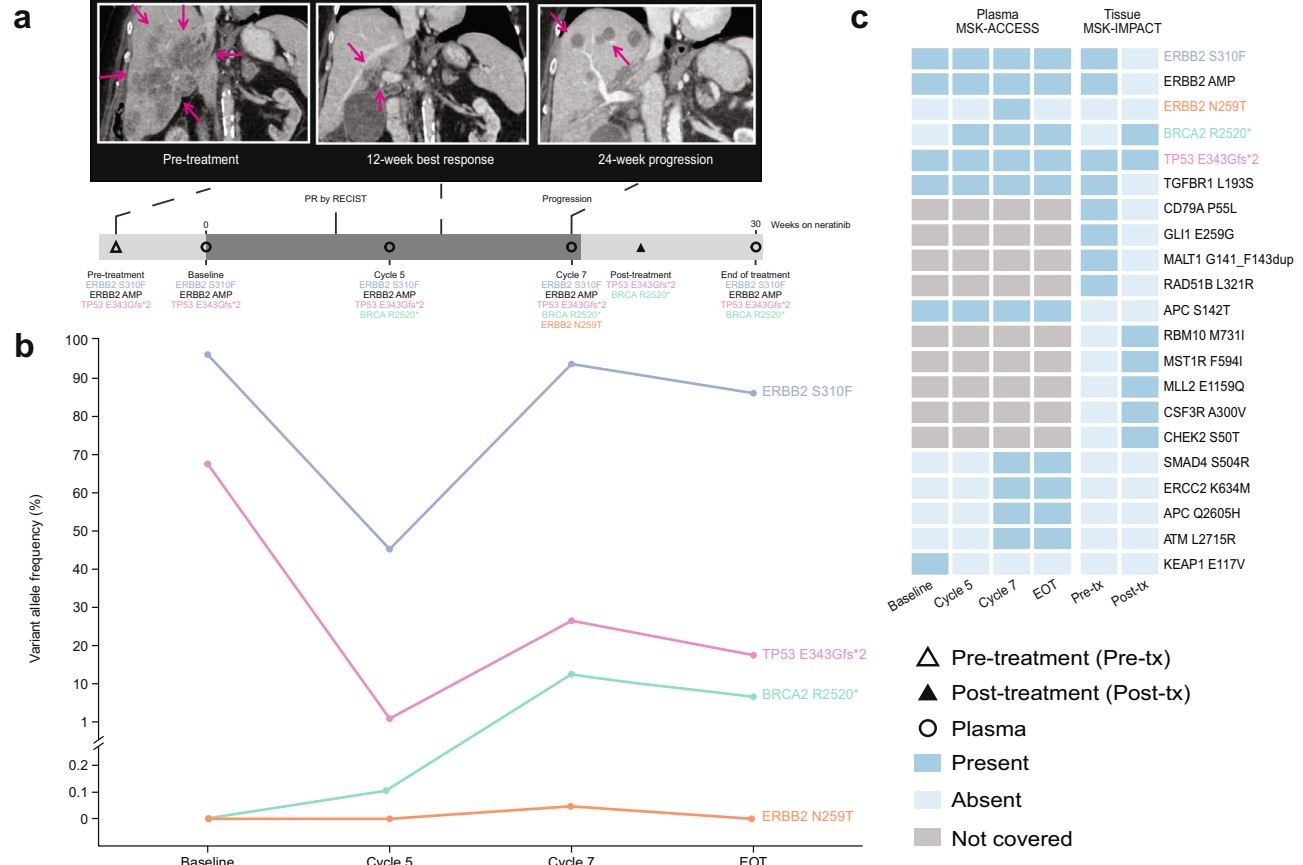

**Fig. 5 | Polyclonal resistance to neratinib.** A patient with adenosquamous carcinoma of the gallbladder harbouring a *HER2* amplified/S310F mutation who had progression of disease on gemcitabine plus cisplatin, FOLFOX, and FOLFIRI achieved a confirmed PR on treatment with neratinib. **a** Cross-sectional and treatment course imaging showing tumour response and progression (purple arrow); **b** serial cfDNA and **c** paired paired-tissue next-generation sequencing. cfDNA cell-free DNA, *EOT* end of treatment, *FOLFIRI* 5-fluorouracil/folinic acid + irinotecan, *FOLFOX* 5-fluorouracil/folinic acid + oxaliplatin, *MSK-ACCESS* Memorial Sloan Kettering-Analysis of Circulating cfDNA to Evaluate Somatic Status, *MSK-IMPACT* Memorial Sloan Kettering-Integrated Mutation Profiling of Actionable Cancer Targets, *PR* partial response, *tx* treatment. Source data are provided as a Source Data file.

disease had progressed on gemcitabine + cisplatin, 5-fluorouracil/folinic acid + oxaliplatin, and 5-fluorouracil/folinic acid + irinotecan (Fig. 5). NGS sequencing of a pre-treatment liver metastasis revealed amplification of the mutant *HER2* S310F allele, a truncating *TP53* mutation, and 19 additional molecular aberrations (six variants of unknown significance and 14 copy-number amplifications). Treatment with neratinib resulted in a PR (best response −48% reduction in target lesions). Biopsy and NGS of a progressing liver lesion revealed loss of *HER2* amplification with reduction in variant allele fraction (VAF) of the original *HER2* S310F mutation (VAF 90.2% pre-treatment vs <0.4% at disease progression). In addition, five mutations and seven copy-number alterations not seen in the pre-treatment sample were detected following disease progression, including a truncating *BRCA2* mutation (R2520*). NGS of peripheral blood before treatment detected *HER2* S310F and *TP53* E343Gfs*2 but not *BRCA2* R2520*. On-treatment plasma sampling revealed reduction in allele frequencies of *HER2* S310F and *TP53* mutations, consistent with response to neratinib treatment; both increased at the time of progression. Of note, the *BRCA2* truncating mutation and an additional *HER2* N259T mutation were detected in the progression plasma sample.

## Discussion
Patients with advanced BTC have limited treatment options and poor overall prognosis, so it is critical to develop new treatments for this disease. This cohort of SUMMIT identified and selected patients with

advanced BTC harbouring infrequent somatic, activating *HER2* mutations and describe their genome-guided treatment with a selective pan-HER TKI[27]. In this BTC cohort, neratinib was safe and exhibited modest antitumour activity, with a confirmed ORR of 16% (95% CI 4.5–36.1%) and CBR of 28.0% (95% CI 12.1–49.4%). The observed antitumour activity as assessed by ORR in the present study is similar to the observed anticancer activity of the currently available second-line cytotoxic chemotherapies (ORR 5–15%) and is in line with the modest ORR of 22% trastuzumab and pertuzumab in HER2 positive BTC[4,5]. These data give credence to the oncogenic role of somatic *HER2* mutations in BTC, provide further support for targeting *HER2* missense mutations as a therapeutic approach, and provide a much-needed benchmark for the operating characteristics of *HER2*-targeted therapy in the rare instance of somatic *HER2*-activating mutations[9]. That said, the lower proportion of patients with tumour shrinkage, short DoR, and the failure of the cohort to meet the prespecified rules that were required for the results to be considered positive argue for both a deeper understanding of ERBB2 oncogene addiction in this rare cancer and the need to focus on strategies such as combination therapy approaches with potential to enhance antitumour activity and improve outcomes.

Objective responses were only seen in patients with cholangiocarcinoma and gallbladder cancers; minimal to no anticancer activity was observed in ampulla of Vater cancers. Interestingly, three of the four ampullary cancers harboured dual *HER2* mutations based on

central sequencing, one of which also had an activating *KRAS* co-mutation. The fourth was not available for central sequencing. In patients with breast cancer, either pre-existent dual *HER2* mutation or *HER2/HER3* co-mutation has been reported to be associated with lack of clinical benefit to neratinib as either a single agent or combined with anti-oestrogen therapy[28]. In addition, one of four patients exhibited intestinal differentiation. Acknowledging the caveats of drawing conclusions given the small sample size, these findings suggest that responses to *HER2*-targeted therapy may differ based on anatomic site, histology, and/or underlying genomic profile, and future studies should seek to address discrete anatomic pathologies as separate cohorts or stratify accordingly[29]. Such findings have also been observed in HER2-positive (amplification or overexpression) BTC tumours treated with monoclonal antibodies; in the MyPathway study, patients with gallbladder cancers tended to have better outcomes than other sub-histologies[18].

Although the genomics for our study were descriptive in nature and objective responses were infrequent, outcomes appeared worse for patients with co-occurring inactivating alterations in cell-cycle regulators *TP53* and *CDKN2A*. A recent, large retrospective study reported that *TP53* and *CDK2NA* alterations have prognostic implications in cholangiocarcinoma, with worse outcomes after adjusting for stage and other known prognostic factors[26]. Furthermore, other reports of HER2-targeted TKIs and HER2-directed antibodies in HER2-positive cancers suggest that *TP53* alterations and other cell-cycle regulators are important genomic modifiers of response and outcome[25,30,31]. Further prospective studies in larger cohorts of patients with BTC harbouring *HER2* mutations will be needed to confirm the predictive and prognostic role of co-occurring *TP53* mutations with or without *CDKN2A* mutations.

Although tumour shrinkage was clearly documented in a subset of patients, responses to neratinib were relatively short-lived, indicating acquisition of resistance (DoR 3.0–4.7 months). We attempted to nominate potential resistance mechanisms though NGS on paired tumour biopsies and on serial cfDNA collection. One of four responders lost *HER2* alterations during progressive sampling, with the emergence of several mutations unique to this resistant clone. At the time of disease progression, analysis of a second metastatic lesion indicated loss of the initial *HER2* copy-number amplification and a marked decrease in VAF of the *HER2* S310F mutation, with emergence of an inactivating *BRCA2* mutation and several other alterations not observed on the pre-treatment sample. In addition, other alterations were observed in cfDNA, including a low-level *HER2* N259T variant of unknown significance, which were not observed in pre-treatment or post-progression tumours. Acquisition of additional *HER2* mutations, whether oncogenic or not, has been reported upon progression with neratinib-containing therapy in *HER2*-mutant breast cancer[28,32]. In a second patient, serial cfDNA analysis did not reveal emergence of an alternative genomic driver, either due to assay detection limitations or via a more complex functional resistance mechanism not measurable by NSG or cfDNA. Given the limited sample size, it is challenging to speculate on generalised mechanisms of acquired resistance in BTC based on these results; preclinical modelling could potentially help inform such pathways.

Neratinib treatment was generally well tolerated in this BTC cohort. The AE profile was comparable with previous reports: diarrhoea and vomiting were the most common all-grade events, and diarrhoea and abdominal pain were the most common grade 3/4 events. No new safety signals were observed. The pattern of AEs observed in the BTC cohort was generally similar to the overall SUMMIT population, in which diarrhoea, nausea, and vomiting were the most common all-grade AEs, affecting 74%, 43%, and 41% of patients, respectively[25].

The study has several notable strengths including: ability to identify rare genomic variants in an uncommon patient population; correlative design that allowed for hypothesis-generating observations

regarding prognostic implications of co-occurring mutations, as well as those related to acquired resistance to neratinib; and implementation of the potential utility of cfDNA in identifying patients with BTC for molecular therapeutics, as the genomic driver of interest was detected in a high proportion of pre-treatment samples and was often concordant with tumour tissue.

Limitations of this study include the small sample size, inability to confirm centrally the oncogenic driver in three of 25 patients, and lack of available pre-treatment tissue for central confirmation in approximately 40% of patients, which hampered correlative analysis. These observations suggest pre-treatment tissue acquisition may be required in future molecularly targeted studies in BTC. In retrospect, as response appeared to differ based on anatomic site, exclusion of ampullary cancers or at least defining anatomic cohorts of sufficient sample size would have led to better estimation of antitumour activity. It is also important to acknowledge, as observed in other studies in patients with BTC[5,6,18,33–36], that a subset of patients progressed rapidly on treatment; eight patients died as a result of their disease within 8 weeks of study initiation. This highlights the complexity of drug development in BTC.

In summary, analysis of the genomically driven, multi-histology SUMMIT trial suggests that selected *HER2*-mutant BTCs are sensitive to inhibition by the pan-HER TKI neratinib. Neratinib was well tolerated and showed antitumour activity in patients with metastatic gallbladder cancer or cholangiocarcinoma harbouring *HER2* mutations. Addition of a second targeted agent prolonged and deepened responses to neratinib in the breast and lung cohorts of SUMMIT[28,37] and a parallel approach could have similar utility for patients with *HER2*-mutant BTC. Likewise, prospective modelling (i.e., I-PREDICT) has illustrated that co-targeting oncogenic drivers with more than one precision medicine might further enhance response[38]. Further studies are needed to evaluate the role of additional agents in enhancing response to neratinib in patients with *HER2*-mutant metastatic BTC.

## Methods
### Study design and participants
This research complies with all relevant ethical regulations regarding the use of human study participants. The study was conducted in accordance with International Conference on Harmonisation Good Clinical Practice guidelines, Declaration of Helsinki, and local regulations. Approval was obtained from institutional review boards at each of the participating institutions (Supplementary Table 4). The SUMMIT protocol, which is available on request to the lead author, is not yet published online as several elements of the SUMMIT trial are ongoing. The protocol will be published on clinicaltrials.gov after completion of the study. Written informed consent was obtained for all patients before performing study-related procedures. Patients were not compensated for taking part in the study.

SUMMIT (NCT01953926) is an open-label, single-arm, multi-cohort, multi-tumour, phase 2, 'basket' trial conducted at 58 centres internationally[25]. Patients were recruited between April 3, 2014, and August 1, 2019 by investigators at each site based on *HER2*-mutation status; 16 sites contributed at least one patient to the BTC cohort.

Inclusion and exclusion criteria are as follows:

### Inclusion criteria

1. Men and women who are ≥18 years old at signing of informed consent.
2. Histologically confirmed cancers in patients with activating *ERBB* mutations and who are refractory to standard therapy or for which standard or curative therapy does not exist or is not considered sufficient or appropriate by the Investigator.
3. At the time of screening, a previously documented mutation: *HER2* mutation in breast, bladder/urinary tract, biliary tract,

colorectal, endometrial, gastroesophageal, lung, ovarian, and any other cancers.

4. At least one measurable lesion, preferably as defined by Response Evaluation Criteria in Solid Tumours (version 1.1)[39].
5. Left ventricular ejection fraction ≥50% measured by multiple-gated acquisition scan or echocardiogram.
6. Eastern Cooperative Oncology Group performance status of 0–2.
7. Female patients with cancers known to secrete β-human chorionic gonadotropin (β-hCG), ie, germinomas, are eligible if the pattern of serum β-hCG is suggestive of the malignancy and the pelvic ultrasound is negative for pregnancy.
8. Men must agree and commit to use a barrier method of contraception while on treatment and for 3 months after the last dose of the investigational product. Women of child-bearing potential must agree and commit to the use of a highly effective double-barrier method of contraception (e.g., a combination of male condom with an intravaginal device such as the cervical cap, diaphragm, or vaginal sponge with spermicide) or a non-hormonal method, from the signing of the informed consent until:
   i. 28 days after the last dose of neratinib monotherapy, or
   ii. 6 months after the last dose of paclitaxel, or
   iii. 1 year after the last dose of fulvestrant.
9. Provide written informed consent to participate in the study and follow the study procedures.

## Exclusion criteria

1. Prior treatment with any *HER2*-directed tyrosine kinase inhibitor (e.g., lapatinib, afatinib, dacomitinib, neratinib) with the exception of patients with non-small cell lung cancer (NSCLC) who may have received afatinib and remain eligible.
2. Not recovered to at least grade 1 or baseline (National Cancer Institute [NCI] Common Terminology Criteria for Adverse Events [CTCAE] version 4.0) from all clinically significant adverse events related to prior therapies (excluding alopecia).
3. Received chemotherapy or biologic therapy ≤2 weeks or five half-lives of the agent used, whichever is shorter, prior to the start of neratinib.
4. Received radiation therapy ≤14 days prior to initiation of investigational product, except primary brain tumour patients.
5. Patients who are receiving any other anticancer agents with the exception of patients on (1) a stable dose of bisphosphonates or denosumab or (2) sex hormone therapy in the case of breast, prostate, or gynaecological cancers.
6. Received prior therapy resulting in a cumulative epirubicin dose >900 mg/m² or cumulative doxorubicin dose >450 mg/m². If another anthracycline or more than one anthracycline has been used, the cumulative dose must not exceed the equivalent of 450 mg/m² doxorubicin.
7. Symptomatic or unstable brain metastases. (Note: Asymptomatic patients with metastatic brain disease who have been on a stable dose of corticosteroids for treatment of brain metastases for at least 14 days are eligible to participate in the study.) Patients with primary central nervous system tumours are eligible.
8. Active uncontrolled cardiac disease, including cardiomyopathy, congestive heart failure (New York Heart Association functional classification of ≥2), unstable angina (symptomatic angina pectoris within the past 180 days that required the initiation of or increase in anti-anginal medication or other intervention), myocardial infarction within 12 months of enrolment, or ventricular arrhythmia (except for benign premature ventricular contractions). For patients with NSCLC, the following are additionally excluded: conduction abnormality requiring a pacemaker; supraventricular and/or nodal arrhythmias not controlled

with medication; valvular disease with documented compromise in cardiac function; symptomatic pericarditis; any history of myocardial infarction documented by elevated cardiac enzymes or persistent regional wall abnormalities on assessment of left ventricular function; any history of documented congestive heart failure and/or cardiomyopathy.

9. Demonstrates a QTc interval >450 ms for men or >470 ms for women or known history of congenital QT prolongation or torsade de pointes.
10. Inadequate bone marrow, renal, or hepatic function as defined on screening laboratory assessments.
11. Uncontrolled concurrent malignancy (early-stage or chronic disease is allowed if not requiring active therapy or intervention and is under control).
12. Active infection or unexplained fever >38.5 °C (101.3 °F).
13. Women who are pregnant, are planning on becoming pregnant, or are breast-feeding.
14. Significant chronic gastrointestinal disorder with diarrhoea as a major symptom (e.g., Crohn's disease, malabsorption, or grade ≥2 NCI CTCAE [version 4.0] diarrhoea of any aetiology at baseline).
15. Clinically active infection with a hepatitis virus.
16. Evidence of significant medical illness, abnormal laboratory finding, or psychiatric illness/social situations that could, in the Investigator's judgement, make the patient inappropriate for this study.
17. Known hypersensitivity to any component of the investigational product, required combination therapy, or loperamide.
18. Unable or unwilling to swallow tablets.
19. Patients bearing certain somatic *HER* mutations, such as those that are subclonal in nature, or resulting in the expression of truncated proteins including alterations that result in a premature stop codon or a change in reading frame (i.e., frameshift mutations) may not be considered for eligibility.
20. Patients with known activating *KRAS* mutations.

## Procedures

Patients received neratinib 240 mg orally daily. One cycle was 28 days or 4 weeks of treatment. Patients received mandatory loperamide prophylaxis. Patients were treated until disease progression, unacceptable toxicity, or consent withdrawal.

Tumour response was assessed locally using RECIST (version 1.1) every 8 weeks by computed tomography or magnetic resonance imaging. AEs were recorded using Common Terminology Criteria for AEs (version 4.0) from consent until day 28 after study treatment discontinuation.

All clinical data were collected under an IRB-approved protocol and store in a encrypted and secure database that underwent regular review and cross referencing in the local electronic medical record. Upon the completion of the protocol, this database was locked. Genomic data were generated for each patient on study and stored in cBioPortal. All data were generated using cBioPortal for Cancer Genomics: MutationMapper (cBioportal front end version 3.7.23)[40,41] (Fig. 3) and ComplexHeatMap (version 2.8.0) package with ggplot2 (3.3.5) and R (4.1.3) (Figs. 4 and 5). All figures were refined using Adobe Illustrator 2021 (25.2.3).

## Exploratory genomic analysis

Archival or pre-treatment formalin-fixed, paraffin-embedded (FFPE) tumour tissue was required for study entry. Plasma was collected before treatment, on treatment (every other cycle), and at treatment discontinuation. Tumour DNA was extracted from FFPE tissue or plasma, and sequenced using Memorial Sloan Kettering-Integrated Mutation Profiling of Actionable Cancer Targets (MSK-IMPACT)[42] or MSK-ACCESS[43]. Custom targeted *HER2* single-gene sequencing was performed in select cases using plasma samples. Somatic alterations were annotated with OncoKB (version date December 24, 2021)[44].

## Outcomes and endpoints

The primary objective of the BTC cohort was to determine the response rate (ORR). Other objectives included determination of: CBR (confirmed CR, PR, or stable disease [SD] for ≥16 weeks within ±7-day visit window); PFS (interval from treatment start to first date on which recurrence, progression, or any-cause death was documented); OS (defined as the interval from start of treatment to death for those who died; for those who did not die, censored at the last known alive time); DoR (time from date measurement criteria were met for CR or PR until first date of documented disease progression); safety; and tolerability. Disease control rate (DCR), which was not a prespecified endpoint, was defined as confirmed CR, PR, or SD for ≥24 weeks. Exploratory correlative objectives included retrospective central confirmation of locally reported *HER2* mutation via NGS on archival or fresh tumour tissue or in cfDNA extracted from plasma, and description of patient outcomes based on pre-treatment genomics, and genomic clonal evolution with treatment via NGS on serial cfDNA.

## Statistical analysis

A Simon two-stage optimal design was used to determine whether neratinib monotherapy had sufficient activity to warrant further development. Early study termination was permitted if data at the first stage indicated treatment was ineffective. Using Simon's optimal two-stage design (significance level 10%, power 80%), a true ORR at 8 weeks of ≤10% was considered unacceptable (null hypothesis) and a true ORR at 8 weeks of minimally 30% (alternative hypothesis) merited further study. In the first stage, enrolment continued until seven patients had completed two neratinib cycles and appropriate activity assessment was completed. If one or more responses, defined as a PR or CR at the first post-baseline assessment, were observed, the cohort was expanded to include 11 additional patients for the second stage. Additional enrolment beyond the first 18 patients was allowed to ensure that at least 18 patients were evaluable for a radiographic response. This led to over-enrolment of the study. If four or more responses were seen in stage 2, the cohort could be expanded to a maximum of 30 patients. As the biliary tract cohort did not meet the criteria for continued enrolment, this cohort was closed to recruitment. The overall SUMMIT study, of which this is a component, is ongoing.

All endpoints were descriptive. No sex- or gender-based analyses were performed as there are no biological data to support a difference in outcome based on sex or gender. Analyses for correlates, which were prespecified, were descriptive. Baseline characteristics, activity, and safety were summarised in the safety analysis set (all patients receiving at least one neratinib dose). The Clopper-Pearson method was used to calculate ORR and CBR 95% CIs. Kaplan–Meier methodology was used to determine PFS estimates with 95% CIs. All statistical analyses were performed using SAS (version 9.4; SAS Institute Inc., Cary, NC, USA) or the survival package (version 3.1–12) from R (version 4.0.2)[45]. This study is registered with ClinicalTrials.gov, NCT01953926 and European Union Drug Regulating Authorities Clinical Trials Database, EudraCT 2013-002872-42.

## Data availability

Puma Biotechnology is committed to sharing clinical trial data and information to help physicians and patients make informed treatment decisions, and to help researchers advance scientific knowledge. The authors declare that the data supporting the findings of this study are available within the article and source data for the figures are provided with this paper. The SUMMIT protocol is not yet published online as several elements of the SUMMIT trial are ongoing but it will be published on ClinicalTrials.gov within 1 year of completion of the study. Raw patient data are under restricted access for privacy reasons. Puma makes patient-level, de-identified data sets, and associated documents available as set forth in Puma's data sharing policy (https://pumabiotechnology.com/data_sharing_policy.html). Requests for study protocol, other study documentation and clinical trial data may be submitted to clinicaltrials@pumabiotechnology.com for consideration. Once approved, a data sharing agreement will be provided for timely access to these data for the time required to perform the analysis. Genomic data have been deposited in the cBioPortal repository (available at https://www.cbioportal.org/study/summary?id=biliary_tract_summit_2022). Source data are provided with this paper.

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

## Acknowledgements

The SUMMIT trial was sponsored/funded by Puma Biotechnology, Inc. Investigators from MSKCC who participated in the trial were also supported in part by a Cancer Center Support Grant (P30 CA008748) and Cycle for Survival. Puma Biotechnology, Inc was involved in the following: study design; data collection, analysis and interpretation of the data; writing of the report; the decision to submit the article for publication. The authors would like to thank all patients and their families for participating in the SUMMIT trial. The authors acknowledge David Hyman (Memorial Sloan Kettering), Richard Bryce (Puma Biotechnology), and Alshad Lalani (Puma Biotechnology) for their important contributions to the original SUMMIT study design, oversight, and interpretation, and Feng Xu (Puma Biotechnology) and Jane Liang (Puma Biotechnology) for statistical and programming support. The authors also thank Lee Miller and Deirdre Carman (Miller Medical Communications Ltd) for medical writing/editing assistance, which was funded by Puma Biotechnology, Inc.

## Author contributions

J.J.H., S.A.P.-P., D.DiP, M.F.B., L.D.E., F.M.-B., D.B.S., and G.K.A.-A. were responsible for the conception and design of the work. J.J.H., S.A.P.-P., R.H.S., J.J.M., J.Z., D.DiP., C.S., S.D.S., M.F.B., L.D.E., F.M.-B., K.J., D.B.S., and G.K.A.-A. were responsible for data analysis. J.J.H., S.A.P.-P., R.H.S., J.M.C., G.I.S., D.I.Q., I.B., V.M., M.B., S.L., I.S., H.P., J.M.F., M.A., S.M.S., CdelaF, C.F., J.Z., DDiP, C.S., S.D.S., M.F.B., L.D.E., F.M.-B., K.J., D.B.S., and G.K.A.-A. were responsible for data acquisition. J.J.H., S.A.P.-P., R.H.S., J.M.C., D.I.Q., I.B., V.M., M.B., S.L., I.S., H.P., J.M.F., M.A., S.M.S., CdelaF, C.F., J.Z., D.DiP., M.F.B., L.D.E., F.M.-B., K.J., D.B.S., and G.K.A.-A. were responsible for data interpretation. J.J.H., R.H.S., J.Z., D.DiP., M.F.B., L.D.E., D.B.S., and G.K.A.-A. were responsible for drafting the work. All authors critically revised the work, approved the final draft of the manuscript, and agree to be accountable for all aspects of the work. J.J.H., R.H.S., J.J.M., J.Z., D.DiP., M.F.B., L.D.E., D.B.S., and G.K.A.-A. verify the underlying data.

## Competing interests

J.J.H. has received institutional research support from Bristol Myers Squibb, Boehringer Ingelheim, CytomX, Calithera, Eli Lilly, Genoscience, Incyte, Loxo, Novartis, Pfizer, Yiviva, and Zymeworks and has consulted for Adaptimmune, Bristol Myers Squibb, CytomX, Eisai, Eli Lilly, Exelixis, Merck, QED, and Zymeworks. S.A.P.-P.: has received clinical trial support (to institution) from AbbVie, Inc., ABM Therapeutics, Inc., Acepodia, Inc., Alkermes, Aminex Therapeutics, Amphivena Therapeutics, Inc., Bio-Marin Pharmaceuticals, Inc., Boehringer Ingelheim, Bristol Myers Squib,

Cerulean Pharma, Inc., Chugai Pharmaceutical Co., Ltd., Curis, Inc., Cyclacel Pharmaceuticals, Daiichi Sankyo, Inc., Eli Lilly, ENB Therapeutics, Five Prime Therapeutics, F-Star Beta Limited, F-Star Therapeutics, Limited, Gene Quantum, Genmab A/S, GlaxoSmithKline, Helix BioPharma Corp., HiberCell, Inc., Immunomedics, Inc., Incyte Corp., Jacobio Pharmaceuticals Co., Ltd., Lytix Biopharma AS, Medimmune, LLC., Medivation, Inc., Merck Sharp and Dohme Corp., Novartis Pharmaceuticals, Pieris Pharmaceuticals, Inc., Pfizer, Principia Biopharma, Inc., Puma Biotechnology, Inc., Rapt Therapeutics, Inc., Seattle Genetics, Silverback Therapeutics, Synlogic Therapeutics, Taiho Oncology, Tesaro, Inc., and TransThera Bio, and research grant NCI/NIH P30CA016672 – Core Grant (CCSG Shared Resources). R.H.S.: no conflicts declared. J.J.M.: no conflicts declared. JMC: has received research support from Merck, AstraZeneca, Esperas, Tesaro, and Bayer, and payments or honoraria from Syros and Blueprint. G.I.S.: has received research funding from Eli Lilly, Merck KGaA/EMD Serono, Merck, and Sierra Oncology. He has served on advisory boards for Pfizer, Eli Lilly, G1 Therapeutics, Merck KGaA/EMD Serono, Sierra Oncology, Bicycle Therapeutics, Fusion Pharmaceuticals, Cybrexa Therapeutics, Astex, Ipsen, Bayer, Angiex, Daiichi Sankyo, Seattle Genetics, Boehringer Ingelheim, ImmunoMet, Asana, Artios, Atrin, Concarlo Holdings, Syros, Zentalis, CytomX Therapeutics, Blueprint Medicines, Kymera Therapeutics, Janssen and Xinthera. In addition, he holds a patent entitled, "Dosage regimen for sapacitabine and seliciclib," also issued to Cyclacel Pharmaceuticals, and a pending patent, entitled, "Compositions and Methods for Predicting Response and Resistance to CDK4/6 Inhibition," together with Liam Cornell. DIQ: has received institutional research support from Merck Sharp and Dohme, Pfizer, Genentech, and Millennium and has consulted for Astellas, Aveo, BMS, Bayer, Genentech/ Roche, EMD Serono, Merck Sharp and Dohme, Pfizer and Seagen, received payments or honoraria from Astellas, Aveo, BMS, Bayer, Genentech/Roche, EMD Serono, Merck Sharp and Dohme, Pfizer and Seagen, undertaken data safety monitoring for Eisai and US Biotest, and has employment with AbbVie. I.B.: has received research funding from Puma Biotechnology Inc for the manuscript (to institution), a personal grant from the Spanish National Health Institute, research funding from Institut Salud Carlos III (personal grant Rio Hortega Contract - CM15/00255) and La Caixa Foundation Institutional grant LCF/PR/CEO7/50610001, research funding from AstraZeneca, Boehringer Ingelheim, Bristol Myers Squibb, Celgene, GlaxoSmithKline, Gliknik, Incyte, ISA Pharmaceuticals, Janssen Oncology, Kura, Merck Serono, Merck Sharp & Dohme, Novartis, Northern Biologics, Orion Pharma, Regeneron, Seattle Genetics, Shattuck Labs, and VCN Biosciences (all institutional research funding to the principal investigator), personal consulting fees from Achilles Therapeutics, Cancer expert Now, eTheRNA Immunotherapies, Merck Sharp & Dohme, and Rakuten Pharma, payments or honoraria from Bristol Myers Squibb, Merck Serono, and Merck Sharp & Dohme (personal fees), support for attending meetings from Merck Serono, and Merck Sharp & Dohme (personal fees), and is an advisory board member of the ESMO Head and Neck Tract, EORTC Head and Neck Group, and Cancer Core Europe Clinical Taskforce. VM: research funding to institution from AbbVie, AceaBio, Adaptimmune, ADC Therapeutics, Aduro, Agenus, Amcure, Amgen, Astellas, AstraZeneca Bayer BeiGene BioInvent International AB, BMS, Boehringer, Boston, Celgene, Daiichi Sankyo, DEBIO-PHARM, Eisai, e-Terapeutics, Exelixis, Forma Therapeutics, Genmab, GSK, Harpoon, Hutchison, Immutep, Incyte, Inovio, Iovance, Janssen, Kyowa Kirin, Lilly, Loxo, MedSir, Menarini, Merck, Merus, Millennium, MSD, Nanobiotix, Nektar, Novartis, Odonate Therapeutics, Pfizer, Pharma Mar, PharmaMar, Principia, PsiOxus, Puma, Regeneron, Rigontec, Roche, Sanofi, Sierra Oncology, Sponsor, Synthon, Taiho, Takeda, Tesaro, Transgene, Turning Point Therapeutics, and Upsher-Smith Laboratories, and personal consulting fees from Roche, Bayer, Janssen, and Basliea. M.B.: reports grants from Adaptimmune, grants from Senhwa Pharmaceuticals, grants from Agios Pharmaceuticals, personal fees from Merck, grants from EMD Serono, grants from Halozyme, grants

from Celgene, grants from Toray, grants from Dicerna, grants from Taiho, grants from Sun Biopharma, grants from Isis Pharmaceuticals, grants from Redhill Pharmaceuticals, grants from Boston Biomed, grants from Basilea, grants from Incyte Pharmaceuticals, grants from Mirna Pharmaceuticals, grants from Medimmune, grants from Bioline, grants from Sillajen, grants from ARIAD, grants from Puma Pharmaceuticals, grants from QED, grants from Novartis Pharmaceuticals, personal fees from ADC Therapeutics, personal fees from Exelixis, personal fees from Inspyr, personal fees from G1 Therapeutics, personal fees from Immunovative, personal fees from OncBioMune, personal fees from Western Oncolytics, personal fees from Lynx Group, personal fees from Genentech, personal fees from HUYA, grants from AstraZeneca, outside the submitted work. S.L.: receives research funding to her institution from Novartis, Bristol Myers Squibb, Merck, Puma Biotechnology, Eli Lilly, Nektar Therapeutics, AstraZeneca, Roche-Genentech and Seattle Genetics. She has acted as consultant (not compensated) to Seattle Genetics, Novartis, Bristol Myers Squibb, Merck, AstraZeneca, Eli Lilly, Pfizer and Roche-Genentech. She has acted as consultant (paid to her institution) to Aduro Biotech, Novartis, GlaxoSmithKline, Roche-Genentech, AstraZeneca, Silverback Therapeutics, G1 Therapeutics, Puma Biotechnologies, Pfizer, Gilead Therapeutics, Seattle Genetics, Daiichi Sankyo, Amunix, Tallac Therapeutics, Eli Lilly and Bristol Myers Squibb. I.S.: has received research support (to institution) from Puma Biotechnology, Roche/Genentech, AstraZeneca, Incyte, Pfizer, Orion Pharma, MSD, Bristol Myers Squibb, Novartis, Loxo Oncology, Amgen, and Genmab, and support for travel and attending meetings from Roche and Novartis. H.P.: writing support for the manuscript from Puma Biotechnology Inc, research grant UM1 CA186689, research support to institution from Adlai Nortye USA, Alpine Immune Sciences, Ambrx, Amgen, Aprea Therapeutics AB, Array BioPharma, Bayer, BeiGene, BJ Bioscience, Bristol Myers Squibb, Daiichi Pharmaceutical, Eli Lilly, Elicio Therapeutics, EMD Serono, Exelixis, Genentech, Gilead Sciences, GlaxoSmithKline, Gossamer Bio, Hoffmann-LaRoche, Hutchison Medi-Pharma, ImmuneOncia Therapeutics, Incyte, Jounce Therapeutics, Mabspace Biosciences, MacroGenics, Medimmune, Medivation, MERCK, Millennium, Mirati Therapeutics, Novartis Pharmaceuticals, Oncologie, Pfizer, PsiOxus Therapeutics, Puma Biotechnology, Regeneron, Pharmaceuticals, RePare Therapeutics, Seattle Genetics, Synermore Biologics, Taiho Pharmaceutical, TopAlliance Biosciences, Turning Point Therapeutics, Vedanta Biosciences, Xencor Inc, honorarium from Medscape, and support for attending meetings from Daiichi Sankyo and Vedanta. JMF: has received research funding from Puma Biotechnology Inc for the manuscript (to institution), and research support from Pfizer, Genentech, Merus, Incyte, and AstraZeneca (all to institution). M.A.: has received research funding from Puma Biotechnology Inc for the manuscript (to institution), research funding from Eli Lilly (to institution), consulting fees from AbbVie (to self), payments or honoraria from Pfizer (to self), Roche (to institution), and AstraZeneca (to institution), and support for attending meetings from Pfizer and AstraZeneca (to self). S.M.S.: has received research funding from Puma Biotechnology, Inc. CdelaF: has received grants or contracts from Pierre Fabre Oncologie; payments or honoraria from Amgen, Bayer, Eisai, Servier, Ipsen, and Lilly; payments for expert testimony from Incyte, MSD, Pierre Fabre Oncologie, Ipsen, Roche, and Daiichi Sankyo; support for attending meetings and/or travel from Amgen, Roche, MSD, and Merck. C.F.: has received research funding from Puma Biotechnology Inc for the manuscript (to institution), research funding from the National Comprehensive Cancer Network Oncology Research Program, Taiho Oncology, and Pfizer Inc (all to institution), and equipment, materials, drugs, medical writing, gifts or other services from Pfizer Inc and Taiho Oncology (both to institution). J.Z.: was an employee of and shareholder in Puma Biotechnology, Inc. D.DiP.: is an employee of and shareholder in Puma Biotechnology, Inc. C.S.: no conflicts declared. S.D.S.: no conflicts declared. M.F.B.: has received research funding from Grail and consulting fees from Eli Lilly and PetDx. L.D.E.: is an employee of and

shareholder in Puma Biotechnology, Inc. F.M.-B.: has received research funding from Puma Biotechnology Inc for the manuscript (to institution), research funding from Aileron Therapeutics, Inc., AstraZeneca, Bayer Healthcare Pharmaceutical, Calithera Biosciences Inc., Curis Inc., CytomX Therapeutics Inc., Daiichi Sankyo Co. Ltd., Debiopharm International, eFFECTOR Therapeutics, Genentech Inc., Guardant Health Inc., Klus Pharma, Takeda Pharmaceutical, Novartis, and Taiho Pharmaceutical Co. (all to institution), consulting fees from AbbVie, Aduro BioTech Inc., Alkermes, AstraZeneca, DebioPharm, eFFECTOR Therapeutics, F. Hoffmann-La Roche Ltd., Genentech Inc., IBM Watson, Infinity Pharmaceuticals, Jackson Laboratory, Kolon Life Science, Lengo Therapeutics, OrigiMed, PACT Pharma, Parexel International, Pfizer Inc., Samsung Bioepis, Seattle Genetics Inc., Tallac Therapeutics, Tyra Biosciences, Xencor, and Zymeworks, an honorarium for speaking engagement from Chugai Biopharmaceuticals, and advisory board membership for Black Diamond, Biovica, Eisai, Immunomedics, Inflection Biosciences, Karyopharm Therapeutics, Loxo Oncology, Mersana Therapeutics, OnCusp Therapeutics, Puma Biotechnology Inc., Seattle Genetics, Silverback Therapeutics, Spectrum Pharmaceuticals, and Zentalis. K.J.: has received research funding from Puma Biotechnology Inc for the manuscript (to institution). D.B.S. has received consulting fees from BridgeBio, Pfizer, Loxo/Lilly Oncology, FORE Therapeutics, Scorpion Therapeutics, and Vividion Therapeutics and owns stock or stock options in Loxo Oncology, Scorpion Therapeutics, and Vividion Therapeutics. GKA-A has received research funding from Puma Biotechnology Inc for the manuscript, institutional research support from Arcus, Agios, AstraZeneca, BioNTech, BMS, Celgene, Flatiron, Genentech/Roche, Genoscience, Incyte, Polaris, Puma, QED, Silenseed, and Yiviva, consulting support form Adicet, Alnylam, AstraZeneca, Autem, Bayer, Beigene, Berry Genomics, Celgene, Cend, CytomX, Eisai, Eli Lilly, Exelixis, Flatiron, Genentech/Roche, Genoscience, Helio, Incyte, Ipsen, Legend Biotech, Merck, Nerviano, QED, Redhill, Rafael, Servier, Silenseed, Sobi, Surface Oncology, Therabionics, Vector, and Yiviva, and a patent (International Patent Application No. PCT/US2014/031545 filed on March 24, 2014, and priority application Serial No.: 61/804,907; Filed: March 25, 2013).

## Additional information

[1]Department of Medicine, Memorial Sloan Kettering Cancer Center, New York, NY, USA. [2]Department of Medicine, Weill Cornell Medical College, New York, NY, USA. [3]Department of Investigational Cancer Therapeutics, The University of Texas, MD Anderson Cancer Center, Houston, TX, USA. [4]Kravis Center for Molecular Oncology, Sloan Kettering Institute, New York, NY, USA. [5]Dana-Farber Cancer Institute, Boston, MA, USA. [6]Keck School of Medicine, USC Norris Cancer Comprehensive Cancer Center, Los Angeles, CA, USA. [7]Medical Oncology Department, Vall d'Hebron University Hospital, Barcelona, Spain. [8]Molecular Therapeutic Research Unit – UITM-La Caixa, Vall d'Hebron Institute of Oncology (VHIO), Barcelona, Spain. [9]START MADRID-FJD, Hospital Fundación Jiménez Díaz, Madrid, Spain. [10]Medical Oncology Department, Mayo Clinic, Scottsdale, AZ, USA. [11]Translational Breast Cancer Genomics and Therapeutics Laboratory, Peter MacCallum Cancer Centre, Melbourne, Australia. [12]Department of Oncology, Rigshospitalet, Copenhagen University Hospital, Copenhagen, Denmark. [13]Division of Oncology, Washington University School of Medicine in St. Louis, St. Louis, MO, USA. [14]Department of Medicine (Oncology), Stanford Cancer Institute, Stanford, CA, USA. [15]Medical Oncology Department, Gustave Roussy, Villejuif, France (currently at: Institut Bergonie, Bordeaux, France. [16]Institute of Oncology, Davidoff Center, Rabin Medical Center, Petach Tiqwa, Israel. [17]The Sackler Faculty of Medicine, Tel Aviv University Tel Aviv, Tel Aviv, Israel. [18]Medical Oncology Department, Centre Léon Bérard, Lyon, France. [19]Division of GI Medicine, Department of Medicine, Roswell Park Comprehensive Cancer Center, Buffalo, NY, USA. [20]Early Phase Clinical Trial Program, Department of Medicine, Roswell Park Comprehensive Cancer Center, Buffalo, NY, USA. [21]Translational Medicine and Diagnostics, Puma Biotechnology Inc, Los Angeles, CA, USA. ✉e-mail: Hardinj1@mskcc.org

