## [Peer Review File · Nature Communications]

Reviewers' Comments:

Reviewer #1:

Remarks to the Author:

This is the clinical report of neratinib in HER2 mutated biliary tract cancer. 25 patients were treated with an ORR of 16%. Detail of emergent resistant mutations is given on one patient.

Reviewer #3:

Remarks to the Author:

This research is an important contribution to the population of individuals with biliary tract cancers. BTC are rare and the basket trial design is an excellent choice to gather information on rare cancers. The paper is well written with solid graphical and tabular displays of results. The power calculation and Type I error rate is correct for the Simon two-stage design. The analysis is, as appropriate, largely descriptive with indication of uncertainty in key outcomes.

Comments:

1) The attached protocol states that once the Stage 2 criteria are met, the cohort can be expanded up to 30 patients. In this trial the Stage 2 criteria was not met. My understanding is that Stage 2 was intended to enroll 18 subjects. However the paper reports on 25, implying that the study continued to enroll beyond 18 patients without evaluating the Stage 2 criteria. Can the authors provide insight into why 25 patients were enrolled rather than 18?

2) Line 102 is confusing. Suggest replace

'The BTC cohort was closed to recruitment when it was determined that the required number of patients did not have an objective response to treatment at Week 8, the first efficacy assessment ' The number of patients with an objective response at week 8 did not achieve the number required to continue enrollment. Hence recruitment concluded with xxx (This refers as well to my comment in (1)). Why did you recruit past 18 subjects.

3) Table 2 could use better formatting. Please indent subcategories to help organize around endpoints. Also include the acronyms in the table for easy connection with the acronyms in the text.

Reviewer #4:

Remarks to the Author:

The authors report, "Targeting HER2 mutant advanced biliary tract cancers with 2 neratinib: results from the SUMMIT 'basket' trial." Per the authors, "SUMMIT is an open-label, single-arm, multi-cohort, phase 2, 'basket' trial of neratinib in patients with solid tumours harbouring oncogenic HER2 somatic mutations (NCT01953926). The primary objective of the BTC cohort was objective response rate (ORR). Among 25 treatment-refractory patients (11 cholangiocarcinoma, 10 gallbladder, 4 ampullary cancers), the ORR was 16%. The most common HER2 mutations were S310F (n=11; 48%) and V777L (n=4; 17%). Outcomes appeared worse for ampullary tumours or those with co-occurring oncogenic TP53 and CDKN2A alterations. Loss of amplified HER2 S310F and acquisition of multiple previously undetected oncogenic co-mutations were identified at progression in one responder."

In Valle J, Wasan H, Palmer DH, et al. "Cisplatin plus gemcitabine versus gemcitabine for biliary tract Cancer" (N Engl J Med 2010;362:1273-81), the ORR was 15.5% versus 26.1%, respectively. Thus, while the authors conclude that "Neratinib demonstrated anti-tumour activity in patients with refractory BTC harbouring HER2 mutations," this is less than exciting given that the ORR is like single agent cisplatin. Please comment and include this data in introduction.

Six of 25 patients with BTC were not evaluable for response. This is nearly a quarter of patients. Why? Why did patients withdraw consent for follow up?

Where the ampullar of Vater tumors pancreatobiliary or intestinal subtype? Is this skewing results with "non-biliary" type tumors?

What is the Disease control rate (DCR) if SD for ≥ 6 months is included?

Using two patients to define resistance mechanisms is underpowered. The rationale for the report is that most series are small. Here, the series is also small, thus contradicting the premise noted in the introduction. Also, is this truly acquired resistance or merely growth of resistant clones attesting to tumor heterogeneity?

Please include protein expression data in the OncoPrint figure (Fig. 3B)

As the authors note, further investigation of combination therapies is warranted. In the I-PREDICT study in Nature Medicine 2019, a molecular matching score was utilized. Given the rare responses seen here, does post-hoc analysis of molecular matching scores correlate with responses? Reviewing Figure 3B, this may be the case. Please analyze.

Further analysis by the same group reported "Comprehensive genomic landscape and precision therapeutic approach in biliary tract cancers." Their findings should be noted in the context of the discussion and higher response rates with precision targeted combination therapies for these cancers.

Another recent references to contextualize includes "ERBB2 Pathway in Biliary Tract Carcinoma: Clinical Implications of a Targetable Pathway" by Jacobi et al. in 2021.

"Limitations of this study include the small sample size, inability to confirm centrally the oncogenic driver in three (12%) of 25 patients, and lack of available pre-treatment tissue for central confirmation in approximately 40% of patients, which hampered correlative analysis." These are significant limitations that weaken the overall impact of the study, especially considering this is a negative overall.

Table 2. ORR of 16.0 is missing a percent sign.

Nature Communications manuscript NCOMMS-22-13385

Responses to reviewers' comments

Reviewer 1

We thank reviewer 1 for the review and comments.

Reviewer #3

This research is an important contribution to the population of individuals with biliary tract cancers. BTC are rare and the basket trial design is an excellent choice to gather information on rare cancers. The paper is well written with solid graphical and tabular displays of results. The power calculation and Type I error rate is correct for the Simon two-stage design. The analysis is, as appropriate, largely descriptive with indication of uncertainty in key outcomes.

We thank the reviewer for these comments on our work.

1) The attached protocol states that once the Stage 2 criteria are met, the cohort can be expanded up to 30 patients. In this trial the Stage 2 criteria was not met. My understanding is that Stage 2 was intended to enroll 18 subjects. However the paper reports on 25, implying that the study continued to enroll beyond 18 patients without evaluating the Stage 2 criteria. Can the authors provide insight into why 25 patients were enrolled rather than 18?

Thank you for the opportunity to clarify. The Reviewer is correct that the second stage of the study over-enrolled. As described in the revised Statistical analysis section of the manuscript (page 16) and in the accompanying protocol, the study required 18 radiographically evaluable patients. The study continued until 18 patients were deemed fully evaluable with at least two scans. As observed in this aggressive disease, and commented on by reviewer #4, a proportion of patients are expected to have clinical decline or complications related to disease. For these reasons, enrolment continued beyond 18 patients in the second stage. In addition, as the study was conducted over multiple global sites and focused on an uncommon patient population with few treatment options, some patients who consented as the study was closing were still allowed to continue if they were otherwise eligible. This was reasonable in that, although the study did not meet the endpoint 8-week ORR, the ORR observed is in line with second- and third-line treatment options for this disease. We have updated the Methods accordingly (page 16).

2) Line 102 is confusing. Suggest replace

'The BTC cohort was closed to recruitment when it was determined that the required number of patients did not have an objective response to treatment at Week 8, the first efficacy assessment ' The number of patients with an objective response at week 8 did not achieve the number required to continue enrollment. Hence recruitment concluded with xxx (This refers as well to my comment in (1)). Why did you recruit past 18 subjects.

We believe this comment has been addressed in the response above. The Methods section of the manuscript has been amended to clarify this point (page 16) and the confusing statement in the results has been removed from the Results section (page 6).

3) Table 2 could use better formatting. Please indent subcategories to help organize around endpoints. Also include the acronyms in the table for easy connection with the acronyms in the text

Table 2 has been reformatted as suggested.

Reviewer #4

The authors report, "Targeting HER2 mutant advanced biliary tract cancers with 2 neratinib: results from the SUMMIT 'basket' trial." Per the authors, "SUMMIT is an open-label, single-arm, multi-cohort, phase 2, 'basket' trial of neratinib in patients with solid tumours harbouring oncogenic HER2 somatic mutations (NCT01953926). The primary objective of the BTC cohort was objective response rate (ORR). Among 25 treatment-refractory patients (11 cholangiocarcinoma, 10 gallbladder, 4 ampullary cancers), the ORR was 16%. The most common HER2 mutations were S310F (n=11; 48%) and V777L (n=4; 17%). Outcomes appeared worse for ampullary tumours or those with co-occurring oncogenic TP53 and CDKN2A alterations. Loss of amplified HER2 S310F and acquisition of multiple previously undetected oncogenic co-mutations were identified at progression in one responder."

In Valle J, Wasan H, Palmer DH, et al. "Cisplatin plus gemcitabine versus gemcitabine for biliary tract Cancer" (N Engl J Med 2010;362:1273-81), the ORR was 15.5% versus 26.1%, respectively. Thus, while the authors conclude that "Neratinib demonstrated anti-tumour activity in patients with refractory BTC harbouring HER2 mutations," this is less than exciting given that the ORR is like single agent cisplatin. Please comment and include this data in introduction.

We thank the reviewer for this suggestion. We have updated the Introduction to include the contemporary front-line TOPAZ-1 study and have included the outcomes of GemCis + durvalumab in the front-line setting. We would like to note, however, that our study was in a heavily pretreated population with a median of two prior lines of treatment. A more appropriate benchmark of activity would be phase 2 or 3 studies conducted in the second line, i.e. ABC-06. We have therefore also added the ORR, PFS, and OS for FOLFOX in the second-line setting based on ABC-06 (5%, 4.0 months, 6.2 months, respectively). Although we acknowledge the hazards of cross-trial comparison in a heterogenous disease type, in comparison with ABC-06 and real-world data, the ORR observed in this study supports the conclusion that neratinib has antitumor activity. We agree that the level of anticancer activity is modest – as are most treatments for this orphan disease – and quantify this as such in the Discussion (page 10). We also acknowledge the need for further development with combination therapy in this space (pages 10, 13).

Six of 25 patients with BTC were not evaluable for response. This is nearly a quarter of patients. Why? Why did patients withdraw consent for follow up?

As noted in Figure 1 flow study, unevaluable patents came off study as a result of clinical deterioration due to disease (n=2), clinical progression (n=3), and death (n=1). This is not surprising and is consistent with published data in the second line and beyond in BTC (See Table below). Review of clinical trials with either FDA-approved agents or those with NCCN guideline-based recommendations indicates that 20–55% of patients with BTC will progress or die in the first 3 months of study in the second line. The Discussion highlights that 25% of patients came off study and notes that this observation is in line with prior data and illustrates the complexity of drug development in BTC (page 13). In order to clarify, the current version of the text reads as follows "It is also important to acknowledge, as observed in other studies in patients with

BTC^{1,2,3,4,5,6,7}, that a subset of patients progressed rapidly on treatment” and now contains the citations below:

Table 1: Progression-Free Survival (PFS) on Prospective Trials or BTC

Study	Agent	N	~3-month PFS (%)	Median PFS (Months)
ABC-06 (Lamarca et al 2021)	FOLFOX	81	66.7	4
NIFTY (Yoo et al 2021)	Nal-Iri	88	65	7.1
ClarIDHy (Abou-Alfa et al 2020)	Ivosidenib	124	45	2.7
Fight-202 (Abou Alfa et al 2020)	Pemigatinib	107	80	6.9
Javle et al (Javle et al 2021)	Infigratinib	108	N/A	7.3
MyPathway (Javle et al 2021)	Trastuzumab + pertuzumab	39	60	4.0
CA209-538 (Klein et al 2020)	Ipilimumab + nivolumab	39	50	2.9

Where the ampullar of Vater tumors pancreatobiliary or intestinal subtype? Is this skewing results with “non-biliary” type tumors?

We appreciate this comment. Of the four patients with ampullary cancer, three were hepatobiliary and one was intestinal type. We have included these data in the demographics table and added to the Discussion (page 11).

What is the Disease control rate (DCR) if SD for ≥ 6 months is included?

The disease control rate is 24.0% (6/25 patients), with a 95% CI of (9.3%, 45.1%). We now include this information in the Results section (page 6).

Using two patients to define resistance mechanisms is underpowered. The rationale for the report is that most series are small. Here, the series is also small, thus contradicting the premise noted in the introduction.

The rationale for the study was to prospectively test HER2 inhibition in the rare subset of patients with *HER2*-mutant BTC and to evaluate for antitumor activity. The primary endpoint of the study was therefore objective response rate and, as noted by reviewed 2, the study was adequately powered for the primary objective. We agree with the reviewer that exploring resistance mechanisms is hypothesis-generating and for this reason is a tertiary/correlative endpoint of the study. As noted in the Discussion sections, the correlative component is hypothesis-generating (page 12). As we have now noted in the Results section (page 8), NGS of serial cfDNA was hypothesis-generating.

Also, is this truly acquired resistance or merely growth of resistant clones attesting to tumor heterogeneity?

This is a good point, and the reviewer is correct in that there is no way to distinguish between the two based on the methods used. We have clarified this point in the manuscript by referring to these mutations as “emergent” rather than acquired (page 11).

Please include protein expression data in the OncoPrint figure (Fig. 3B)
Although we appreciate this suggestion by the reviewer, protein expression was not evaluated in these samples and this information cannot be included.
As the authors note, further investigation of combination therapies is warranted. In the I-PREDICT study in Nature Medicine 2019, a molecular matching score was utilized. Given the rare responses seen here, does post-hoc analysis of molecular matching scores correlate with responses? Reviewing Figure 3B, this may be the case. Please analyze.
Although we appreciate the value of the I-PREDICT study, we would note as per the reviewer, that the sample size for correlatives is small and hypothesis-generating. Given the size, we are concerned with multiple comparisons in a sample of 25 patients. We have elected to briefly discuss the relevant and suggested paper in the Discussion (page 13).
Further analysis by the same group reported “Comprehensive genomic landscape and precision therapeutic approach in biliary tract cancers.” Their findings should be noted in the context of the discussion and higher response rates with precision targeted combination therapies for these cancers.
We thank the reviewer for this comment. In addition to the comment above, we have revised the text accordingly and added this citation (Discussion page 13).
Another recent references to contextualize includes “ERBB2 Pathway in Biliary Tract Carcinoma: Clinical Implications of a Targetable Pathway” by Jacobi et al. in 2021.
We thank the reviewer for this comment. We have added this reference to the Introduction (page 4).
“Limitations of this study include the small sample size, inability to confirm centrally the oncogenic driver in three (12%) of 25 patients, and lack of available pre-treatment tissue for central confirmation in approximately 40% of patients, which hampered correlative analysis.” These are significant limitations that weaken the overall impact of the study, especially considering this is a negative overall.
We agree with the reviewer that these are limitations of the study, as noted in the Discussion text.
Table 2. ORR of 16.0 is missing a percent sign.
Table 2 has been amended as suggested.

Editorial comments

For studies involving human research participants- The Reporting Summary should include whether sex and/or gender was considered in the study design and whether sex and/or gender of participants was determined based on self-report or assigned (and methodology used).

Sex was collected based on self-report and the study did not differentiate between sex and/or gender. Sex and/or gender was not considered in the study design as there are no biologic data to support a difference in outcome based on sex and/or gender. We have updated the manuscript and reporting summary accordingly.

Data should be reported disaggregated for sex and gender where this information has been collected and consent has been obtained for reporting and sharing individual-level data; disaggregated numbers for individual experiments must be provided in the source data as appropriate whereas overall numbers may be provided in the Nature Portfolio Reporting Summary.

The is not applicable based on our response above.

In addition, please note that if sex- and gender-based analyses have been performed a priori, results should be reported regardless of positive or negative outcome. We discourage conducting post hoc sex- and gender-based analysis if the study design is insufficient (for example, low sample size) to enable meaningful conclusions.

If no sex- and gender-based analyses have been performed, please indicate the reasons for the lack of these analyses in the Reporting Summary

There was no analysis performed based on sex or gender.

All Nature Communications manuscripts must include a “Data Availability” section after the Methods section but before the References. If any of the data can only be shared on request or are subject to restrictions, please specify the reasons and explain how, when, and by whom the data can be accessed.

The manuscript contains a Data Availability statement as required.

In compliance with Puma’s Clinical Trial Data Sharing Policy, the datasets generated during and/or analysed during the current study are available from the corresponding author on reasonable request and all genomic data will be available at the [cBioPortal.org](https://www.ncbi.nlm.nih.gov/cBioPortal/).

The authors declare that the data supporting the findings of this study are available within the article. Qualified researchers and study participants may submit requests for other study documentation and clinical trial data to clinicaltrials@pumabiotechnology.com for consideration.

The authors declare that the data supporting the findings of this study are available within the article. Qualified researchers and study participants may submit requests for other study documentation and clinical trial data to clinicaltrials@pumabiotechnology.com for consideration.

Genomic data will be available at the cBioPortal.org.

To maximise the reproducibility of research data, we strongly encourage you to provide a file containing the raw data underlying the following types of display items:

- Any reported means/averages in box plots, bar charts, and tables
- Dot plots/scatter plots, especially when there are overlapping points
- Line graphs

The data should be provided in a single Excel file with data for each figure/table in a separate sheet, or in multiple labelled files within a zipped folder. Name this file or folder 'Source Data', and include a brief description in your cover letter. The "Data Availability" section should also include the statement "Source data are provided with this paper."

All the clinical data are reported in the manuscript body and supplement as described above. The details of the sequencing will be reported in [cBioPortal](https://cBioPortal.org).

References

1. Kim RD, *et al.* A phase 2 multi-institutional study of nivolumab for patients with advanced refractory biliary tract cancer. *JAMA Oncol.* **6**, 888-894 (2020).
2. Lamarca A, *et al.* Second-line FOLFOX chemotherapy versus active symptom control for advanced biliary tract cancer (ABC-06): a phase 3, open-label, randomised, controlled trial. *Lancet Oncol.* **22**, 690-701 (2021).
3. Yoo C, *et al.* Liposomal irinotecan plus fluorouracil and leucovorin versus fluorouracil and leucovorin for metastatic biliary tract cancer after progression on gemcitabine plus cisplatin (NIFTY): a multicentre, open-label, randomised, phase 2b study. *Lancet Oncol.* **22**, 1560-1572 (2021).
4. Abou-Alfa GK, *et al.* Ivosidenib in IDH1-mutant, chemotherapy-refractory cholangiocarcinoma (ClarIDHy): a multicentre, randomised, double-blind, placebo-controlled, phase 3 study. *Lancet Oncol.* **21**, 796-807 (2020).
5. Javle M, *et al.* Pertuzumab and trastuzumab for HER2-positive, metastatic biliary tract cancer (MyPathway): a multicentre, open-label, phase 2a, multiple basket study. *Lancet Oncol.* **22**, 1290-1300 (2021).
6. Klein O, *et al.* Evaluation of combination nivolumab and ipilimumab immunotherapy in patients with advanced biliary tract cancers: Subgroup analysis of a phase 2 nonrandomized clinical trial. *JAMA oncology* **6**, 1405-1409 (2020).
7. Javle M, *et al.* Updated results from a phase II study of infigratinib (BGJ398), a selective pan-FGFR kinase inhibitor, in patients with previously treated advanced cholangiocarcinoma containing FGFR2 fusions. *Ann. Oncol.* **29**, viii720 (2018).

Reviewers' Comments:

Reviewer #3:

Remarks to the Author:

1. The authors have offered a clear response to my critique in the response letter. However I think more clarity could be helpful in lines 341-343 regarding the description of how the conduct of the study was consistent with the enrolled sample sizes. The manuscript states (lines 341-343):

Additional enrolment beyond the first 18 patients was allowed to assure all patients were evaluable for a radiographic response. This led to over-enrolment of the study. If four or more responses were seen in stage 2, the cohort could be expanded to a maximum of 30 patients. Biostatistics for the primary analysis were powered by Simon two-stage design.

Do they want to say that:

Additional enrolment beyond the first 18 patients was allowed to [assure all] ENSURE THAT AT LEAST 18 patients were evaluable for a radiographic response.

The statement 'Biostatistics for the primary analysis were powered by Simon two-stage design.' does not make sense and does seem necessary. Earlier the authors indicated that they used the two-stage design, and now they have clarified why there were more than 18 patients enrolled as specified by the design. I think that's all that's necessary.

2. The paper describes the ORR, but on line 321 ORR_first is defined as the outcome. Should this be ORR?

Reviewer #4:

Remarks to the Author:

No further comments/suggestions.

Reviewer comments

Reviewer 3
1. The authors have offered a clear response to my critique in the response letter. However I think more clarity could be helpful in lines 341-343 regarding the description of how the conduct of the study was consistent with the enrolled sample sizes. The manuscript states (lines 341-343): Additional enrolment beyond the first 18 patients was allowed to assure all patients were evaluable for a radiographic response. This led to over-enrolment of the study. If four or more responses were seen in stage 2, the cohort could be expanded to a maximum of 30 patients. Biostatistics for the primary analysis were powered by Simon two-stage design. Do they want to say that: Additional enrolment beyond the first 18 patients was allowed to [assure all] ENSURE THAT AT LEAST 18 patients were evaluable for a radiographic response.
We thank the reviewer for this suggestion and have changed the text accordingly (page 16)
The statement 'Biostatistics for the primary analysis were powered by Simon two-stage design.' does not make sense and does seem necessary. Earlier the authors indicated that they used the two-stage design, and now they have clarified why there were more than 18 patients enrolled as specified by the design. I think that's all that's necessary.
This sentence has been deleted as suggested by the reviewer (page 16)
2. The paper describes the ORR, but on line 321 ORR first is defined as the outcome. Should this be ORR?
This sentence has been amended as suggested (page 15)

Editorial office comments

In order to accept your paper, we require the following:
• A revised author checklist describing your response to our editorial requests (attached).
The author checklist has been completed
• A separate point-by-point response to the reviewers' comments, reproduced verbatim
Done
• The final version of your manuscript as a Word or LaTeX file, with all changes highlighted in the text and any tables prepared using the table menu in Word or the table environment in LaTeX
This has been done as requested
• If using LaTeX, please use numerical references only for citations, and include the references within the manuscript file itself. If you wish to use BibTeX, please copy the reference list from the .bbl file, paste it into the main manuscript .tex file, and delete the associated \bibliography and \bibliographystyle commands.
Not applicable
The complete author list provided in the manuscript file, which must match that given on our manuscript tracking system. The author list in the main manuscript file will be used during typesetting of your article
This has been checked and confirmed
Production-quality versions of each figure as a separate file containing all panels. To ensure the swift processing of your paper, please provide the highest quality versions of your images and when combining different figure parts into one file for layout, use a vector-based application such as Adobe Illustrator or Microsoft Powerpoint. We recommend .ai, .eps, .pdf, .ppt. Figures divided

into panels should be labelled with a lower-case, boldface 'a', 'b', etc. in the top left-hand corner. If resolution is not of sufficient quality, production of your paper will be held whilst replacement files are obtained. For detailed guidance on figure preparation, see https://www.nature.com/documents/aj-artworkguidelines.pdf - Please note that we do not modify the text in figures to conform to style during the production process. Please ensure that your figures are presented accurately and adhere to the guidance provided.
This has been done as requested
Any updated checklists that verify compliance with our research ethics and data reporting standards in PDF format
The EPC and Reporting Summary forms have been updated as requested
The final version of the Supplementary Information in one PDF file
The Supplementary Information file has been uploaded in this format
Any Supplementary Movie, Audio, Data and Software submitted as separate files. Supplementary Data and Source Data must be provided as .xls, .xlsx or .zip files, while Supplementary Software must be supplied as .zip files. ** Please note that we do not edit Supplementary Information files; they must be finalised prior to acceptance of the paper. **
Not applicable
If you wish, an interesting image (but not an illustration or schematic) for consideration as a Featured Image on the Nature Communications homepage. The file should be 1200x675 pixels in RGB format and should be uploaded as a Related Manuscript File. In addition to our home page, we may also use this image (with credit) in other journal-specific promotional material.
We appreciate this suggestion but do not wish to include another figure as a Featured Image
Completed and signed copies of our Multimedia License to Publish (LTP) for any Featured Image suggestions (please use one form for each image and give a scientific description of the image in the 'title' field; do not use "Featured Image" as a title): http://www.nature.com/documents/snl-multimedia-ltp.docx
Not applicable
Open access: Nature Communications is a fully open access journal. Articles are made freely accessible on publication under a CC BY license (Creative Commons Attribution 4.0 International License). This license allows maximum dissemination and re-use of open access materials and is preferred by many research funding bodies.
Agreed
ORCID: Nature Communications is committed to improving transparency in authorship. As part of our efforts in this direction, we are now requesting that all authors identified as 'corresponding author' create and link their Open Researcher and Contributor Identifier (ORCID) with their account on the Manuscript Tracking System (MTS) prior to acceptance. ORCID helps the scientific community achieve unambiguous attribution of all scholarly contributions. For more information please visit http://www.springernature.com/orcid For all corresponding authors listed on the manuscript, please follow the instructions in the link below to link your ORCID to your account on our MTS before submitting the final version of the manuscript. If you do not yet have an ORCID you will be able to create one in minutes. https://www.springernature.com/gp/researchers/orcid/orcid-for-nature-research IMPORTANT: All authors identified as 'corresponding author' on the manuscript must follow these instructions. Non-corresponding authors do not have to link their ORCIDs but are encouraged to do so. Please note that it will not be possible to add/modify ORCIDs at proof. Thus, if they wish to have their ORCID added to the paper they must also follow the above procedure prior to acceptance.
ORCIDs have been provided where available